# Identification of Dietary Phytochemicals Capable of Enhancing the Autophagy Flux in HeLa and Caco-2 Human Cell Lines

**DOI:** 10.3390/antiox9121193

**Published:** 2020-11-27

**Authors:** Kohta Ohnishi, Satoshi Yano, Moe Fujimoto, Maiko Sakai, Erika Harumoto, Airi Furuichi, Masashi Masuda, Hirokazu Ohminami, Hisami Yamanaka-Okumura, Taichi Hara, Yutaka Taketani

**Affiliations:** 1Department of Clinical Nutrition and Food Management, Institute of Biomedical Sciences, Tokushima University Graduate School, 3-18-15 Kuramoto-cho, Tokushima 770-8503, Japan; moexile@gmail.com (M.F.); c201841001@tokushima-u.ac.jp (M.S.); c201702013@tokushima-u.ac.jp (E.H.); c201702017@tokushima-u.ac.jp (A.F.); masuda.masashi@tokushima-u.ac.jp (M.M.); ohminami@tokushima-u.ac.jp (H.O.); okumurah@tokushima-u.ac.jp (H.Y.-O.); 2Laboratory of Food and Life Science, Faculty of Human Sciences, Waseda University, 2-579-15 Mikajima, Tokorozawa 359-1192, Japan; s.yano3@kurenai.waseda.jp

**Keywords:** phytochemicals, autophagy flux, structure–activity relationship, mTOR, p62

## Abstract

Autophagy is a major degradation system for intracellular macromolecules. Its decline with age or obesity is related to the onset and development of various intractable diseases. Although dietary phytochemicals are expected to enhance autophagy for preventive medicine, few studies have addressed their effects on the autophagy flux, which is the focus of the current study. Herein, 67 dietary phytochemicals were screened using a green fluorescent protein (GFP)-microtubule-associated protein light chain 3 (LC3)-red fluorescent protein (RFP)-LC3ΔG probe for the quantitative assessment of autophagic degradation. Among them, isorhamnetin, chrysoeriol, 2,2′,4′-trihydroxychalcone, and zerumbone enhanced the autophagy flux in HeLa cells. Meanwhile, analysis of the structure–activity relationships indicated that the 3′-methoxy-4′-hydroxy group on the B-ring in the flavone skeleton and an *ortho*-phenolic group on the chalcone B-ring were crucial for phytochemicals activities. These active compounds were also effective in colon carcinoma Caco-2 cells, and some of them increased the expression of p62 protein, a typical substrate of autophagic proteolysis, indicating that phytochemicals impact p62 levels in autophagy-dependent and/or -independent manners. In addition, these compounds were characterized by distinct modes of action. While isorhamnetin and chrysoeriol enhanced autophagy in an mTOR signaling-dependent manner, the actions of 2,2′,4′-trihydroxychalcone and zerumbone were independent of mTOR signaling. Hence, these dietary phytochemicals may prove effective as potential preventive or therapeutic strategies for lifestyle-related diseases.

## 1. Introduction

Autophagy is one of the principal systems for the degradation of intracellular components, including proteins [1], lipids [2], and nucleotides [3], as well as cytosolic aggregates and organelles [4]. This bulk degradation system is activated in response to a variety of cellular stress stimuli, such as nutritional deprivation [5], oxidative stress [6,7], bacterial infection [8], and accumulation of damaged organelles [9,10], and allows adaptation to these stresses. In the past decades, autophagy was reported to be crucial for the maintenance of cellular homeostasis in mammals and to prevent the onset of various diseases, such as neurodegeneration [11], cancer [12], diabetes [13], glomerulosclerosis [14], muscle atrophy [15], Crohn’s disease [16], and others. Importantly, autophagic activity declines with age in several organisms including mammals [17,18], which is associated with increased expression of Rubicon, a negative regulator of autophagy [19,20]. Based on these observations, it is strongly suggested that the pharmacological regulation of autophagy may serve as a potential preventive or therapeutic strategy for aging and lifestyle-related diseases [21,22].

The autophagic process begins with the biogenesis of autophagosomes, which are double-membrane vesicles surrounding cytoplasmic substrates. Multiple autophagy-related (ATG) proteins are involved in this step [23]; in particular, the lipidated form of an ATG8 homolog protein, microtubule-associated protein light chain 3-II (LC3-II), is specifically localized on the membrane of autophagosomes [24]. Subsequently, autophagosomes fuse with lysosomes, the contents of which, including more than 60 types of acid hydrolases, combine with the substrates inside autophagosomes, such as p62 protein [25], and ultimately causes their degradation. As for the evaluation of autophagic activity, the expression level of LC3-II, which is closely correlated with the number of intracellular autophagosomes, is regarded as an indicator of autophagy. However, data on LC3-II expression are difficult to interpret, as it becomes upregulated following increased autophagosome formation and lysosomal dysfunction, leading to the accumulation of undegraded autophagosomes [26]. Meanwhile, since p62 is associated with intracellular ubiquitinated proteins destined for autophagic degradation [25], its decreased expression is considered a hallmark of autophagic activation. However, many studies have shown that oxidative [27] or inflammatory [28] stimuli dramatically increase p62 expression at the mRNA level through the transactivation of NRF2 [6] or CHOP [29]. It was, therefore, concluded that p62 expression alone cannot serve as a determinant of autophagic activity [30], especially in the case of oxidative or chemical stresses. Due to the possible ambiguity in the interpretation of LC3-II and p62 data, the importance of an accurate quantification of the overall autophagic degradation has been recently recognized [30].

In 2016, a novel methodology for measuring autophagy flux, based on a green fluorescent protein (GFP)-LC3-red fluorescent protein (RFP)-LC3ΔG probe, was established [31,32]. When expressed in cells, endogenous ATG4 protease cleaves this probe, producing equimolar amounts of GFP-LC3 and RFP-LC3ΔG. GFP-LC3 is then degraded through autophagy, while RFP-LC3ΔG remains in the cytosol to serve as an internal control. Therefore, the GFP/RFP ratio inversely correlates with cellular autophagic activity. This system allows for highly accurate determination of the autophagy flux. Using this probe system, Kaizuka et al. performed a high-throughput screening of an approved drug library, which identified new chemical inducers and inhibitors of autophagy [31].

Several signaling pathways are also involved in the autophagic process. For instance, amino acid starvation serves as a major trigger for autophagy in eukaryotes through mTORC1 inactivation [5,33]. In the presence of amino acids, activated mTORC1 on the lysosomal surface inactivates ULK1 by phosphorylating it at Ser757, thereby suppressing autophagy initiation [34]. Additionally, AMPK activation caused by energetic, oxidative, and genomic stresses initiates the autophagic process, in cooperation with mTORC1–ULK1 signaling [35]. Whereas such nutrients- or energy-sensing mechanisms are well known to be involved in autophagy regulation, little is known regarding the relationship between dietary non-nutrients and autophagy. Many studies have shown that certain dietary phytochemicals, primarily consisting of plant secondary metabolites, are potent modulators of physiological functions in mammals. In particular, the suitability of polyphenolic compounds enriched in foods, such as flavanols [36] in vegetables and fruits, catechins [37] in green tea, and stilbenoids [38] in red wine, has been explored, and their antioxidant, anti-inflammatory, and lifespan-extending properties have been demonstrated. However, the effects of these natural compounds on cellular autophagy remain to be fully elucidated. Certain polyphenolic phytochemicals, such as resveratrol [39] and epigallocatechin gallate [40], have been reported to modulate autophagic activity in mammalian cell lines and nematodes. However, the assay system used for measuring autophagic activity might not be suitable, since some of the natural compounds have been found to affect the KEAP1–NRF2 system [41], which is a master regulator of cellular responses against oxidative stresses, leading to p62 upregulation [6]. Our previous report indeed showed that zerumbone, an electrophilic sesquiterpene derived from shampoo ginger, induced p62 expression even though it activated the protein quality control systems, including autophagy [42]. Furthermore, p62 protein has been reported to directly interact with KEAP1, resulting in NRF2 transactivation [7], as well as in a robust increase in p62 expression through a positive feedback mechanism [6]. Considering such complexity in interpreting p62 expression in the case of treatment with phytochemicals, the effects of dietary non-nutrients on autophagic activity should be re-assessed by the quantitative methodology based on autophagy flux.

In this study, we prepared a screening system capable of quantifying the cellular autophagy flux using HeLa cells expressing a GFP-LC3-RFP-LC3ΔG probe and Operetta, a high-content imaging system. We then screened 67 types of phytochemicals and identified 4 dietary components that enhanced the autophagy flux. As for active compounds, autophagy-enhancing activities and modes of action in the intestine were further investigated using the Caco-2 human colon carcinoma cell line. The results of this study may provide fundamental insights into the development of possible preventive strategies for various diseases, such as neurodegenerative and lifestyle-related diseases associated with autophagy dysfunction.

## 2. Materials and Methods

### 2.1. Reagents

Dulbecco’s modified Eagle’s medium (DMEM; 4.5 g/L glucose; 08458-16), a penicillin–streptomycin mixed solution (Stabilized; 09367-34), bovine serum albumin (01281-26), a protease inhibitor cocktail (25955-11), and a phosphatase inhibitor cocktail (04080-11) were obtained from Nacalai Tesque Inc. (Kyoto, Japan). Puromycin dihydrochloride (AG-CN2-0078) was purchased from Adipogen Life Sciences (San Diego, CA, USA). Fetal bovine serum (FBS; 10270-106) and FluoroBrite™ DMEM (A1896701) were obtained from Thermo Scientific Inc. (Waltham, MA, USA). MEM non-essential amino acid solution (×100) (139-15651) and the Protein Assay BCA Kit (297-73101) were purchased from FUJIFILM Wako Pure Chemical Corporation (Osaka, Japan). Bafilomycin A1 (14005) and Torin1 (10997) were obtained from Cayman Chemical Co. (Ann Arbor, MI, USA). Lipofectamine™ 2000 Transfection Reagent (11668) was obtained from Invitrogen (Carlsbad, CA, USA). pMRX-IP-GFP-LC3-RFP-LC3ΔG (RDB14600) and pMRX-IP-GFP-LC3-RFP (RDB14601) were purchased from RIKEN BRC DNA BANK (Tsukuba, Japan).

The mouse monoclonal antibody against GFP (E-AB-20086) and the rabbit polyclonal antibody against RFP (E-AB-20052) were from Elabscience Inc. (Houston, TX, USA). Phosho-p70 S6K (#97596) and 4EBP1 (#9644) were purchased from Cell Signaling Technology Inc. (Beverly, MA, USA). The mouse monoclonal antibody against SQSTM1 (D3; sc-28359) and the mouse polyclonal antibody against β-actin (C4; sc-4778) were purchased from Santa Cruz Biotechnology Inc. (Dallas, TX, USA). HRP-conjugated antibodies against mouse or rabbit IgG (AP124P or AP124P) were purchased from Millipore Inc. (Billerica, MA, USA).

### 2.2. Phytochemicals

The phytochemicals were obtained as follows. Chrysoeriol (1104S), 6-*O*-Me-luteolin (520-11-6), 7,4′-dihydroxyflavone (1259), datiscetin (1141S), eriodictyol (0056), flavonol (1026), homoeriodictyol (1118S), isorhamnetin-3G (1228), isorhamnetin (1120S), kaempferol-3G (1243G), luteolin-7G (1126S), quercetin-3G (0074), quercetagetin (1030), rhamnetin (1136S), tamarixetin (1140S), and tricetin (1335S) were obtained from Extrasynthese (Genay, France); 2,2′,4′-trihydroxychalcone (T502), 4,2′,5′-trihydroxychalcone (22-314), and gossypetin (G500) were purchased from INDOFINE Chemical Company Inc. (Hillsborough, NJ, USA). Flavone (16012-31), morin (23416-31), and pyrogallol (29703-52) were obtained from Nacalai Tesque Inc. (Kyoto, Japan). EGCG (E4268), hyperoside (00180585-25MG), and phloroglucinol (79330-25G) were purchased from Sigma-Aldrich (St. Louis, MO, USA); 1,2,4-trihydroxybenzene, daidzin, piceid, rutin, and tangeretin were purchased from Tokyo Chemical Industry Co., Ltd. (Tokyo, Japan); 3,4-dihydroxybenzoic acid, 3,5-dihydroxybenzoic acid, cyanidin, hesperetin, and sudachitin were purchased from FUJIFILM Wako Pure Chemical Corporation. Zerumbone was obtained from Adipogen Life Sciences (San Diego, CA, USA). Other phytochemicals were from natural compound libraries (S990043-NAT1/NAT2) purchased from Sigma-Aldrich (St. Louis, MO, USA).

### 2.3. Cell Culture

HeLa cells, a human cervix epithelioid carcinoma cell line, and Caco-2 cells, a human epithelial colorectal adenocarcinoma cell line, were obtained from the American Type Culture Collection. HeLa cells were cultured in DMEM supplemented with 10% FBS and 1% penicillin–streptomycin, and Caco-2 cells were cultured in DMEM supplemented with 10% FBS, 1% penicillin–streptomycin, and 1% non-essential amino acids at 37 °C in a humidified 5% CO_2_ atmosphere.

### 2.4. Generation of Transgenic HeLa Cells Expressing GFP-LC3-RFP-LC3ΔG

HeLa cells were transfected with pMRX-IP-GFP-LC3-RFP-LC3ΔG using Lipofectamine 2000, and puromycin-resistant cells were selectively cultured for 2 weeks. The cells were then subjected to single-cell sorting using a JSAN cell sorter (Bay Bioscience, Hyogo, Japan) to yield a single clone that strongly emitted both GFP and RFP fluorescence. The FL1 filter (λ ex: 488 nm, λ em: 525 nm) and FL7 filter (λ ex: 586 nm, λ em: 600 nm) were used for the detection of GFP and RFP, respectively.

### 2.5. Determination of Cellular Autophagy Flux in HeLa Cells

Transgenic and wild-type HeLa cells were co-cultured in black-wall clear-bottom 96-well plates. After treatment with phytochemicals (2–50 µM), Torin1 (250 nM), or bafilomycin A1 (100 nM) for 6–24 h, the cells were washed with FluoroBrite™ DMEM and imaged using an Operetta high-content imaging system at 40× magnification with the following settings: λ ex: 460–490 nm, λ em: 500–550 nm for GFP and λ ex: 530–560 nm, λ em: 570–650 nm for RFP. Cell shapes were individually recognized by their digital phase-contrast images, and RFP-positive and -negative cells were classified as transgenic and wild-type cells, respectively. Intracellular fluorescence intensities of GFP and RFP were quantified using dedicated imaging software (Harmony 4.5; PerkinElmer, Waltham, MA, USA). To calculate the net fluorescence intensities of the intracellular GFP-LC3-RFP-LC3ΔG probe, the average fluorescence intensity of wild-type cells was subtracted from that of transgenic cells. The GFP/RFP ratio was then calculated as an index of autophagy flux.

### 2.6. Determination of Cellular Autophagy Flux in Transgenic Caco-2 Cells Expressing GFP-LC3-RFP

HEK293FT cells were transiently co-transfected with pMRX-IP-GFP-LC3-RFP, pCG-VSV-G, and pCG-gag/pol using FuGENE HD. After culturing for 72 h, the medium containing virus particles was collected. Caco-2 cells were incubated with the virus-containing medium and polybrene (8 μg/mL) for 48 h, after which the uninfected cells were removed by puromycin treatment (5 μg/mL). After treatment with phytochemicals, the cells were collected, and the fluorescence intensity was measured using Cellometer^®^ Vision (Nexcelom Bioscience LLC, Lawrence, MA, USA). FCS Express4 (De Novo Software, Glendale, CA, USA) was used for quantitative analysis.

### 2.7. Western Blotting

Cells were lysed with Tris–Triton buffer containing 50 mM Tris-HCl (pH 7.4), 150 mM NaCl, 1 mM EDTA, 1% TritonX-100, and a proteinase inhibitor cocktail. The cell lysates were centrifuged at 15,000× *g* for 15 min, and the supernatants were collected. Protein concentration was determined using a Protein Assay BCA Kit, and equal amounts of protein lysates were subjected to SDS-PAGE and electrophoretically transferred to a PVDF membrane (Merck KGaA, Darmstadt, Germany). The membrane was first blocked with TBST buffer (500 mM NaCl, 20 mM Tris-HCl (pH 7.4), and 0.1% Tween 20) containing 5% nonfat dry milk and then incubated with specific primary antibodies overnight at 4 °C and finally with HRP-conjugated secondary antibodies for 1 h. The bound antibodies were detected using the ECL system, and the intensity of protein bands was quantified using FUSION SOLO S (Vilber Lourmat, Marne-la-Vallée, France).

### 2.8. Statistical Analysis

The data are expressed as the mean ± SEM and were statistically analyzed by unpaired Student’s *t*-test for comparison between two independent groups or one-way analysis of variance (ANOVA) with post hoc Tukey–Kramer test for multiple comparisons. The threshold for statistical significance was set at *p* < 0.05.

## 3. Results

### 3.1. Establishment of a Screening System for Chemical Modulators of the Autophagy Flux

We first established an assay system suitable for the screening of dietary phytochemicals able to enhance the autophagy flux. To this end, HeLa cells stably expressing a GFP-LC3-RFP-LC3ΔG probe [31] were prepared. Fluoroimaging of intracellular GFP and RFP indicated that treatment of transgenic cells with Torin1, an mTOR inhibitor, notably decreased GFP, but not RFP, fluorescence, whereas treatment with bafilomycin A1, an inhibitor of lysosomal acidification, resulted in an increase in GFP-positive puncta (Figure 1A). Similar results were obtained from the analysis of GFP-LC3 and RFP-LC3ΔG protein expression by western blotting (Figure 1B). These experiments demonstrated that the generated transgenic cells were a suitable system for the evaluation of the autophagic flux.

Since some phytochemicals are known to emit strong autofluorescence, to obtain a reliable and highly quantitative screening system, the autofluorescence values of cells and tested phytochemicals were subtracted. Co-cultured transgenic and wild-type HeLa cells were imaged in the same field of view, and the fluorescence intensities of GFP and RFP in each cell type were individually quantified using Operetta, a high-content imaging system (Figure 1C). This enabled us to estimate the autofluorescence of wild-type cells and, therefore, to perform an accurate quantification of the net autophagy flux in transgenic HeLa cells.

### 3.2. Identification of Dietary Phytochemicals Enhancing Cellular Autophagy Flux

The described screening system was applied to evaluate the effects of 67 kinds of dietary phytochemicals, including flavonols, flavanonols, flavanones, flavanols, anthocyanidins, isoflavones, flavones, chalcones, stilbenes, phenolic acids, among others, on autophagy flux. All compounds were used at 10 μM for 12 h, as some of them exhibited severe cytotoxicity at higher concentrations (data not shown). Nearly all tested phytochemicals did not cause remarkable changes in the GFP/RFP ratio. Notably, none of the compounds showed obvious inhibitory effects on the autophagy flux as compared to bafilomycin A1 (Figure 2A). However, we found that in transgenic cells, the GFP/RFP ratio tended to decrease after treatment with four specific phytochemicals (Figure 2B), i.e., isorhamnetin, a flavonol contained in onion and apple, chrysoeriol, a flavone derived from celery and bergamot, 2,2′,4′-trihydroxychalcone, contained in Spanish licorice, and zerumbone, a sesquiterpene derived from shampoo ginger. Our previous studies have demonstrated the ability of electrophilic zerumbone to modify thiol groups in cellular proteins, which causes transient proteo-stress in cultured cells [43,44]. Intriguingly, this terpenoid has also been demonstrated to activate protein quality control systems, including autophagy, through p62 upregulation in Hepa1c1c7 murine hepatoma cells [42]. The effects of the other three flavonoid compounds on the autophagy flux are unclear, although some studies have suggested that isorhamnetin and chrysoeriol modulate cellular autophagy [45,46].

To validate the effects of the three flavonoids, triplicate experiments were conducted, and the results demonstrated that cell treatment with isorhamnetin or chrysoeriol, at concentrations of 10 μM or higher, significantly enhanced the autophagic flux. Meanwhile, 2,2′,4′-trihydroxychalcone was effective at concentrations as low as 2 μM (Figure 3A); however, chrysoeriol and 2,2′,4′-trihydroxychalcone induced cytotoxicity at 50 μM (data not shown). In addition, whereas the effects of isorhamnetin and chrysoeriol were only observed after 12–24 h of treatment, that of 2,2′,4′-trihydroxychalcone was detectable after shorter times (Figure 3B). These results demonstrate that isorhamnetin, chrysoeriol, and 2,2′,4′-trihydroxychalcone have the potential to enhance the autophagy flux in human cell lines and suggest that 2,2′,4′-trihydroxychalcone may function by an alternative mechanism to those of isorhamnetin and chrysoeriol.

### 3.3. Analysis of Structure–Activity Relationship of Flavonoids as Modulators of the Autophagy Flux

To investigate the relationship between the chemical structures of phytochemicals and their impact on autophagy, we compared the effects of active flavonoids with those of corresponding analogs. First, the activity of isorhamnetin, containing a mono-methoxylated B-ring structure, was compared to that of four other flavonol compounds with slightly different B-ring moieties (quercetin, kaempferol, myricetin, and tamarixetin; Figure 4B). Interestingly, treatment with these isorhamnetin analogs did not significantly affect the autophagy flux (Figure 4A), indicating that the chemical structure of the 3′-methoxy-4′-hydroxy group on the B-ring moiety in isorhamnetin was critical for its autophagy-promoting activity. Note that the chemical structure of chrysoeriol is similar to that of isorhamnetin, save for the lack of the 3-hydroxy group on the C-ring moiety (Figure 4D). We then evaluated the activities of other flavones with different B-ring structures (luteolin, apigenin, tricetin, and diosmetin; Figure 4D) and found that the chrysoeriol analogs were inactive (Figure 4C). In addition, we investigated the significance of the A- and C-ring structures of chrysoeriol by comparing the activity of chrysoeriol with those of sudachitin, a flavone bearing a 6,8-dimethoxylated A-ring, and homoeriodictyol, a flavanone with a saturated C-ring (Figure 4F). As shown in Figure 4E, both compounds failed to promote the autophagy flux. The data shown in Figure 4A–F indicate that the chemical structure of chrysoeriol was crucial for its effect on the autophagy flux and that structural modifications, except for hydroxylation at the 3-position on the C-ring, severely impaired its activity. Regarding chalcones, isoliquiritigenin has the same A-ring structure, a 2′,4′-di-hydroxy group, as 2,2′,4′-trihydroxychalcone; however, the mono-hydroxylated position on the B-ring differs (Figure 4H). The fact that this analog had negligible effects on the autophagy flux (Figure 4G) suggests that the phenolic hydroxyl group at the *ortho* position of the B-ring plays a critical role in chalcone-induced autophagy modulation.

### 3.4. Effect of Active Phytochemicals on p62 Protein Expression

The gastrointestinal tract is considered to be a crucial organ that is exposed to high concentrations of dietary phytochemicals [47]. Thus, we generated human colon carcinoma Caco-2 cells stably expressing the GFP-LC3-RFP probe to investigate the effects of phytochemicals on autophagy flux in human intestinal cells. Similar to what we observed in HeLa cells, treatment of Caco-2 cells with Torin1 significantly decreased the GFP/RFP ratio (Figure 5A), indicating enhanced autophagy flux. Of note, the four active phytochemicals identified by our screening (Figure 2), i.e., zerumbone, isorhamnetin, chrysoeriol, and 2,2′,4′-trihydroxychalcone, also showed significant effects on Caco-2 cells (Figure 5A).

We also examined the protein expression of p62, which is regarded as a typical substrate of autophagic proteolysis [25]. Treatment of Caco-2 cells with Torin1 resulted in decreased p62 expression, likely due to autophagy activation (Figure 5B,C). However, zerumbone, chrysoeriol, and 2,2′,4′-trihydroxychalcone tended to upregulate p62 expression (Figure 5B,C), despite their potent activity as activators of the autophagy flux (Figure 5A). We reasoned that these contradictory results may be due to a phytochemical-induced increase in p62 mRNA transcription. Indeed, we previously reported that treatment with zerumbone significantly upregulated p62 mRNA levels in Hepa1c1c7 cells [42], possibly involving NRF2 transactivation [48]. The same apparent inconsistency was observed in HeLa cells treated with various flavonoid compounds (Figure 4A–H and Appendix A). These results indicate that evaluating the effects of phytochemicals on cellular autophagy may be a complex task. They also emphasize the importance of accurately assessing the autophagy flux.

### 3.5. mTOR-Dependent and -Independent Mechanisms of Autophagy Induction by Phytochemicals in Caco-2 Cells

Finally, to investigate the molecular mechanisms underlying autophagy activation by these phytochemicals, we assessed the combinatorial effects with Torin1. Interestingly, significant additive effects were observed in cells co-treated with Torin1 and zerumbone or 2,2′,4′-trihydroxychalcone (Figure 6A). Since Torin1 is a selective inhibitor of mTOR, these compounds may modulate the autophagy flux, at least in part, by mTOR-independent mechanisms. Meanwhile, the combination of Torin1 with isorhamnetin or chrysoeriol did not result in additive or synergistic induction of the autophagy flux (Figure 6A), indicating that these compounds promote autophagy in an mTOR-dependent manner.

We further investigated whether mTORC1 signaling was influenced by these phytochemicals using an anti-phospho-p70 S6K (for detecting Thr389 phosphorylation) and an anti-4EBP1 antibody (for assessing 4EBP1phosphorylation via band shift) to evaluate the phosphorylation status of the mTORC1 substrates p70 S6K and 4EBP1. Torin1 was observed to inhibit phosphorylation of S6K (Figure 6B,C) and 4EBP1 (Figure 6B), confirming its potent inhibitory effect on mTORC1 signaling. Of the four selected phytochemicals, only 2,2′,4′-trihydroxychalcone significantly suppressed mTORC1 activity, whereas the others had no significant effects (Figure 6B,C). Taken together, these results suggest that different mechanisms are responsible for the autophagy-promoting effects of these phytochemicals. Notably, although isorhamnetin and chrysoeriol failed to suppress mTORC1 activity, mTOR-related signaling is considered to be critical for their stimulatory effects on autophagy, implying that these flavonoids might act on mTORC2 signaling. Meanwhile, zerumbone and 2,2′,4′-trihydroxychalcone appeared to enhance autophagy in an mTOR-independent manner, although the effects of 2,2′,4′-trihydroxychalcone might be partly dependent on its suppression of mTORC1 signaling.

## 4. Discussion

In the present study, we prepared an assay system for the quantitative assessment of the autophagy flux using a GFP-LC3-RFP-LC3ΔG probe [31] and an Operetta high-content imaging system. The screening of 67 dietary phytochemicals led to the identification of 4 active compounds capable of enhancing the autophagy flux in both HeLa and Caco-2 cells. Our data showed that certain active phytochemicals upregulated p62 expression, emphasizing the importance of directly assessing the autophagy flux rather than the level of p62 expression when evaluating the effects of natural compounds. Interestingly, the active compounds identified in this study were found to act by distinct mechanisms, which were dependent or independent on mTOR-related signaling.

During the optimization of the screening conditions, we noticed that cells treated with several phytochemicals possessed marked autofluorescence even after washing. For example, some flavonols and chalcones emitted strong green and red fluorescence, respectively (data not shown). To avoid this interference, we simultaneously imaged wild-type and transgenic cells in the same field of view (Figure 1C). The determination of the fluorescence intensities of each cell type enabled us to subtract the signal caused by autofluorescence from the measurements and, therefore, to obtain the net GFP/RFP ratio emitted from the probe. This step was essential for an accurate determination of the autophagy flux.

Of the 67 screened compounds, 4 dietary phytochemicals, i.e., isorhamnetin, chrysoeriol, 2,2′,4′-trihydroxychalcone, and zerumbone, enhanced the autophagy flux in HeLa and Caco-2 cells, whereas the others exhibited negligible effects on autophagy (Figure 2A). Meanwhile, some of the phytochemicals determined to be inactive in our assay were previously reported to modulate autophagy in in vitro and in vivo models. For instance, resveratrol, a polyphenolic stilbene abundant in red wine, was shown to activate the autophagy flux at 100 μM in a similar system based on probe-expressing HeLa cells [31]. Intriguingly, resveratrol has been shown to promote longevity in nematodes through SIRT1-dependent induction of autophagy [49]. Moreover, other polyphenols such as apigenin [50], butein [51], epigallocatechin gallate [40], and quercetin [52] have also been suggested to affect autophagy. However, few studies have analyzed their specific effect on the autophagy flux. As shown in Figure 5 and Appendix A, we found that the level of p62 protein did not always reflect the extent of cellular autophagy flux (Table 1). A large number of studies have shown that phytochemicals of a specific category trigger the transactivation of NRF2 [53] or CHOP [54], which promotes p62 expression [6,29]. Furthermore, electrophilic phytochemicals, such as isothiocyanates, and redox-sensitive polyphenols are known to inactivate KEAP1, a negative regulator of NRF2, by modifying or oxidizing its reactive thiol groups [55]. Our previous studies showed that zerumbone directly modifies the thiol groups of cellular proteins, causing KEAP1 inactivation and ER stress and leading to NRF2 and CHOP transactivation [42]. These findings suggest that adopting the level of p62 expression as a measure of autophagic activity may be misleading, particularly in the case of treatment with phytochemicals.

Relative GFP/RFP ratios (Figure 5A) and expression levels of p62 protein (Figure 5C) in Caco-2 cells treated with Torin1 or phytochemicals for 4 h. 

Analysis of the structure–activity relationships indicated that the pro-autophagic effects of flavonoids were strictly dependent on specific chemical features (Figure 4). For example, the 3′-methoxy-4′-hydroxy group on the B-ring was crucial for flavonol and flavone activities, and A- and C-ring moieties also appeared to be important. Moreover, the phenolic hydroxyl group at the *ortho* position of the B-ring of chalcone played a critical role in its activity. Since flavonoids with high redox-sensitive properties, such as quercetin and luteolin—bearing catechol groups—or isoliquiritigenin—possessing a phenolic hydroxyl group at the *para* position—showed relatively low activities, it was concluded that autophagic activation by phytochemicals was independent of their antioxidant properties. Isorhamnetin and chrysoeriol, which are structurally similar, enhanced the autophagy flux in an mTOR-dependent manner; however, they did not suppress mTORC1 activity (Figure 6). Moreover, we recently demonstrated that isorhamnetin promotes lysosomal proteolysis in an mTORC1-independent manner in J774.1 murine macrophage-like cells [56]. These findings suggest that the activation of autophagy-lysosome proteolysis by isorhamnetin and chrysoeriol may occur by the selective modulation of mTORC2 signaling. We found that 2,2′,4′-trihydroxychalcone exerted a strong suppressive effect on mTORC1 signaling, which was likely related to its pro-autophagic activity. However, it also exhibited additive pro-autophagic effects in the presence of Torin1, suggesting the additional involvement of an as yet undefined mTOR-independent mechanism. Meanwhile, although zerumbone has been reported to suppress both mTORC1 and mTORC2 signals in oral squamous cell carcinoma at high concentrations [57], it did not affect mTORC1 activity in this study (Figure 6C). This finding, along with the fact that zerumbone strongly enhanced the autophagy flux even in Torin1-treated cells, suggests that this compound activates a unique mechanism that is completely independent of mTOR signaling. In our previous report, we found that zerumbone induces p62, a cargo receptor of aggrephagy [58] implicated in protein quality control in Hepa1c1c7 cells [42]. Consistent with these findings, some phytochemicals, including zerumbone, increased the expression of p62 protein in Caco-2 cells (Figure 5). Since many polyphenolic flavonoids are NRF2 activators [55], p62 upregulation might serve as a common key event in the pro-autophagic effects of phytochemicals (Figure 7).

It should also be noted that the in vivo bioavailability of orally administered phytochemicals is generally quite poor. Several studies have shown that most dietary phytochemicals are metabolized by phase I and/or II detoxification machineries, primarily in the small intestine and liver, and are subsequently rapidly excreted in the urine, which explains their low concentrations in plasma and peripheral tissues [47,59]. Therefore, the gastrointestinal tract, containing high concentrations of unmetabolized food ingredients, might be an important target organ for functional food factors [60,61].

In this study, we demonstrated the remarkable pro-autophagic activity of four phytochemicals in the Caco-2 human colon adenocarcinoma cell line. These dietary components may prove suitable for application in new preventive and therapeutic strategies for aging and lifestyle-related diseases. However, further in vivo studies investigating the effect of the oral administration of these active phytochemicals are required to verify these findings and future applications.

## 5. Conclusions

We identified four dietary phytochemicals, namely, isorhamnetin, chrysoeriol, 2,2′,4′-trihydroxychalcone, and zerumbone, as autophagic inducers in human cell lines. The 3′-methoxy-4′-hydroxy group on the B-ring in the flavone skeleton and a phenolic hydroxyl group in the *ortho* position on the B-ring of chalcone appeared to be crucial for the pro-autophagic effects of these compounds. Moreover, measurement of the autophagy flux was found to be necessary to accurately assess the effects of phytochemicals, as only some of them affected p62 expression levels. Interestingly, while the action of isorhamnetin and chrysoeriol depended on mTOR signaling, the effects of 2,2′,4′-trihydroxychalcone and zerumbone were at least partially independent of this pathway. These findings could serve to inform the development of novel preventive strategies against various diseases, such as neurodegenerative and lifestyle-related diseases associated with autophagy dysfunction.

## Figures and Tables

**Figure 1 antioxidants-09-01193-f001:**
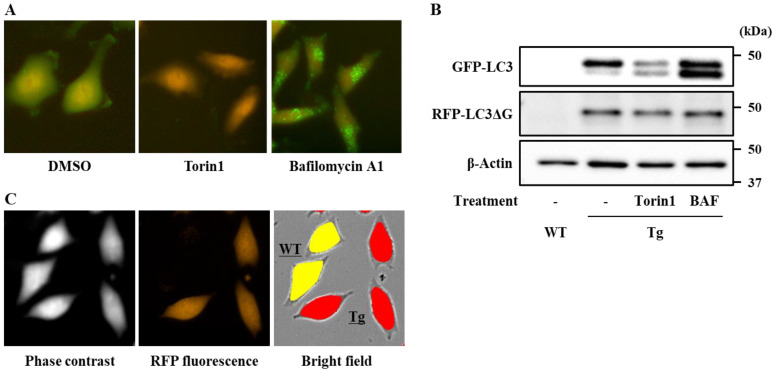
Establishment of an assay system for detecting cellular autophagy flux. HeLa cells stably expressing green fluorescent protein (GFP)-microtubule-associated protein light chain 3 (LC3)-red fluorescent protein (RFP)-LC3ΔG were treated with Torin1 (250 nM) or bafilomycin A1 (BAF; 100 nM) for 12 h (**A**,**B**). Probe expression was detected by fluoroimaging (**A**) and western blotting using antibodies against GFP or RFP (**B**); β-Actin was used as an internal control. Using an Operetta high-content imaging system, co-cultured transgenic and wild-type HeLa cells were distinguished by digital phase-contrast and RFP fluorescence (yellow, wild-type; red, transgenic); the net fluorescence derived from the probe was calculated to obtain the GFP/RFP ratio (**C**).

**Figure 2 antioxidants-09-01193-f002:**
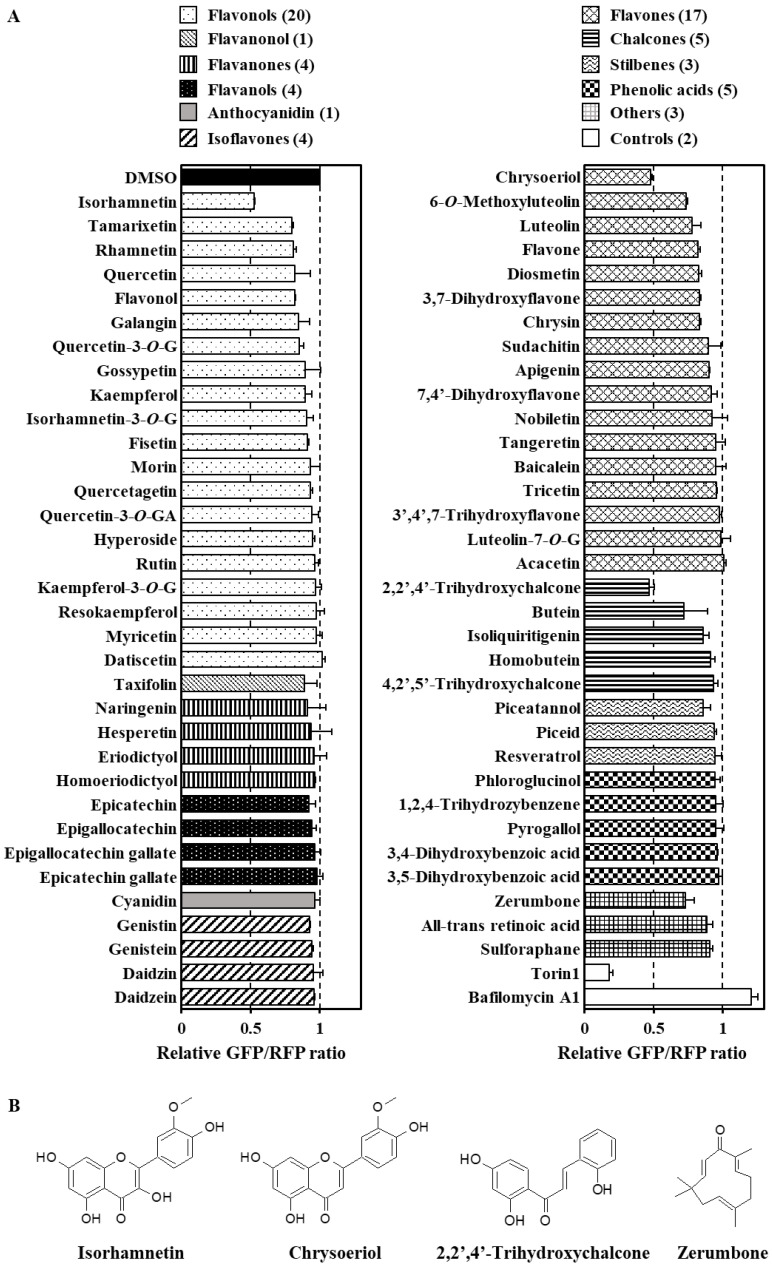
Four dietary phytochemicals, identified using the screening assay, that enhance the autophagy flux. Co-cultured transgenic and wild-type HeLa cells were treated with each phytochemical (10 μM), Torin1 (250 nM), or bafilomycin A1 (100 nM) for 12 h. The cells were then visualized using an Operetta high-content imaging system at 40× magnification to obtain GFP/RFP ratios (n = 2) (–*O*–G, –*O* glucoside; –*O*–GA, –*O*–glucuronide) (**A**). The chemical structures of phytochemicals enhancing the autophagy flux are shown (**B**).

**Figure 3 antioxidants-09-01193-f003:**
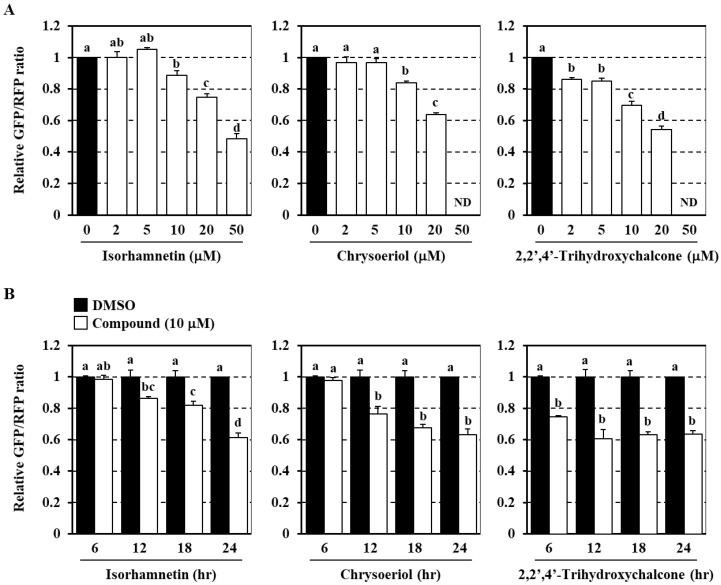
Dose- and time-dependent effects of flavonoids on cellular autophagy flux. Co-cultured transgenic and wild-type HeLa cells were treated with each flavonoid at 2–50 μM for 12 h (**A**) or at 10 μM for 6–24 h (**B**). The cells were then imaged using an Operetta high-content imaging system at 40× magnification to obtain GFP/RFP ratios. Data represent the mean ± SEM of triplicate measurements. The different letters indicate significant differences as determined using the Tukey–Kramer test (*p* < 0.05). ND, not detected.

**Figure 4 antioxidants-09-01193-f004:**
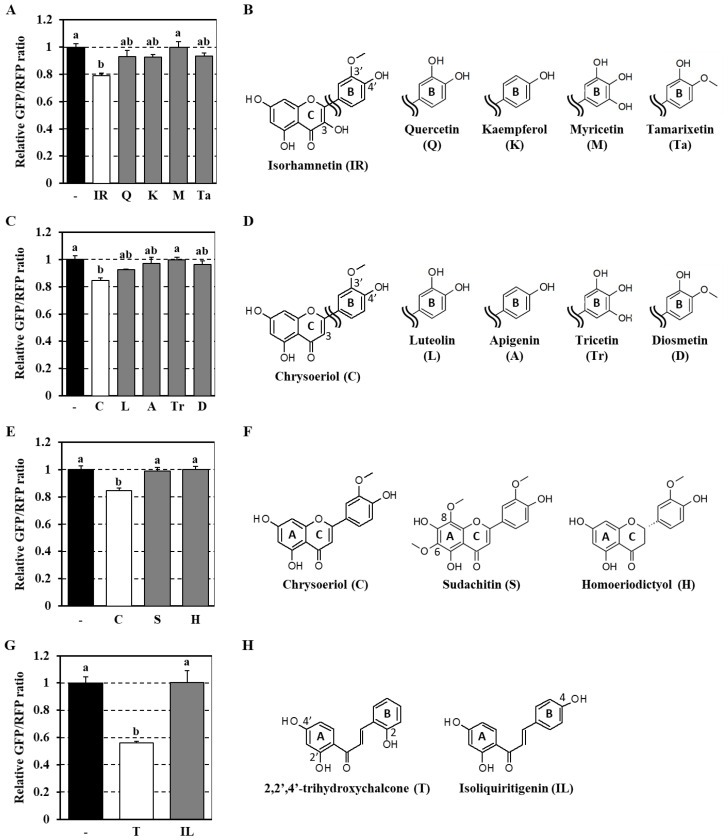
Structure–activity relationships of flavonoids as regulators of the autophagy flux. Co-cultured transgenic and wild-type HeLa cells were treated with each flavonoid (10 μM) for 12 h (**A**,**C**,**E**,**G**). The cells were then visualized using an Operetta high-content imaging system at 40× magnification to obtain GFP/RFP ratios. Data represent the mean ± SEM of triplicate measurements. Different letters indicate significant differences, as assessed by the Tukey–Kramer test (*p* < 0.05). The chemical structures of the tested phytochemicals are shown (**B**,**D**,**F**,**H**).

**Figure 5 antioxidants-09-01193-f005:**
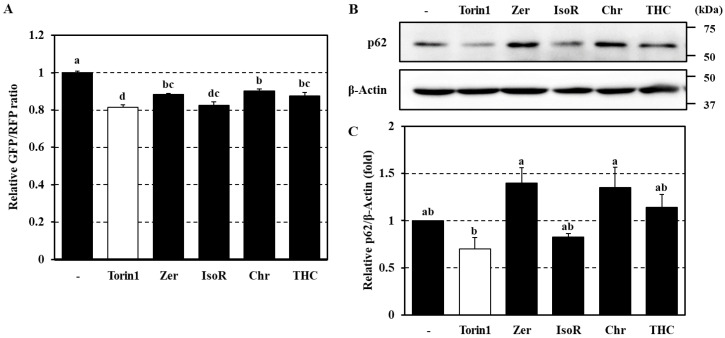
Effects of active phytochemicals on p62 protein expressions in Caco-2 cells. Caco-2 cells were treated with Torin1 (1 μM) or each flavonoid (20 μM) for 4 h. To determine the autophagy flux, intracellular fluorescence intensities were measured using a Cellometer^®^ Vision instrument to obtain GFP/RFP ratios (**A**). p62 protein level was assessed by western blotting. β-actin was used as an internal control (**B**,**C**). Data represent the mean ± SEM of triplicate measurements. Different letters indicate significant differences as assessed by the Tukey–Kramer test (*p* < 0.05) (**C**). Zer, zerumbone; IsoR, isorhamnetin; Chr, chrysoeriol; THC, 2,2′,4′-trihydroxychalcone.

**Figure 6 antioxidants-09-01193-f006:**
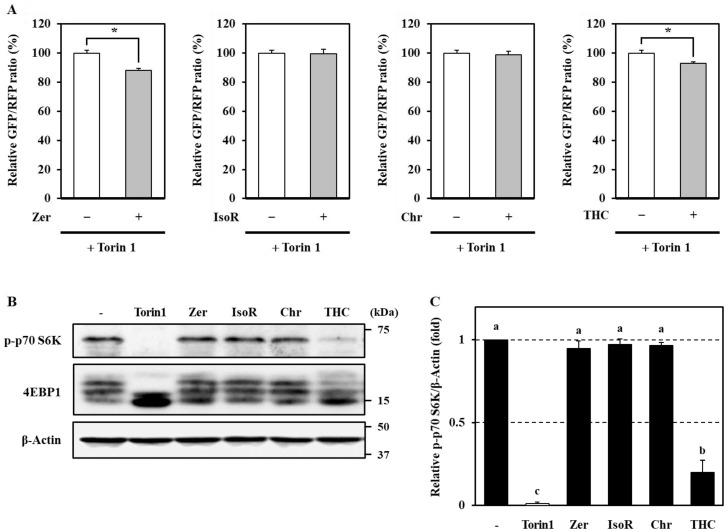
mTOR signaling contribution to the autophagy-promoting effects of phytochemicals. Transgenic Caco-2 cells were treated with Torin1 (1 μM) and/or each flavonoid (20 μM) for 4 h. To determine the autophagy flux, intracellular fluorescence intensities were measured using a Cellometer^®^ Vision instrument to obtain GFP/RFP ratios (**A**). mTORC1 activity was assessed by determining the level of phospho-p70 S6K and 4EBP1 band shifts. β-Actin was used as an internal control (**B**,**C**). Data represent the mean ± SEM of triplicate measurements. Statistical significance was assessed by unpaired *t*-test (* *p* < 0.05) (**A**). The different letters indicate significant differences as determined using the Tukey–Kramer test (*p* < 0.05) (**C**). Zer, zerumbone; IsoR, isorhamnetin; Chr, chrysoeriol; THC, 2,2′,4′-trihydroxychalcone.

**Figure 7 antioxidants-09-01193-f007:**
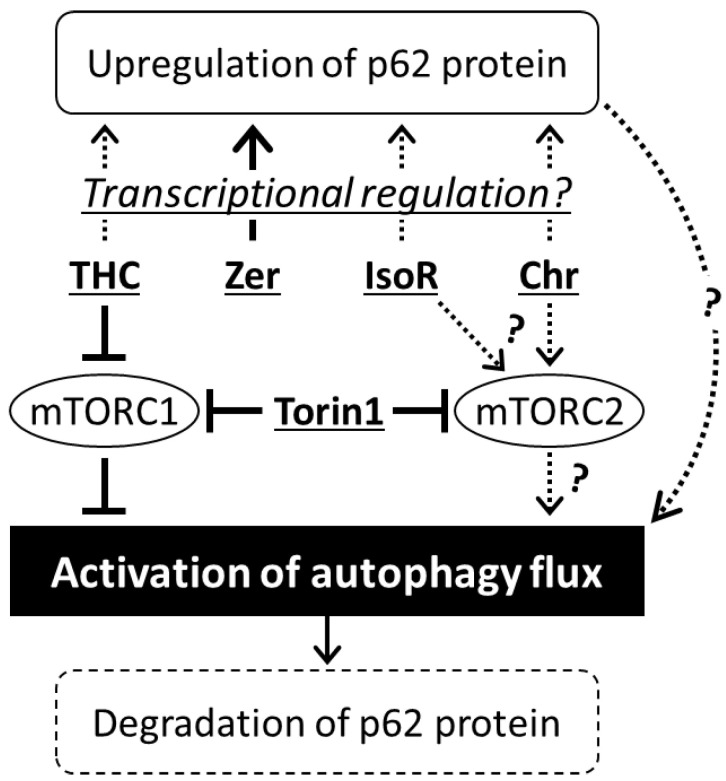
Possible molecular mechanisms underlying autophagy activation by phytochemicals. THC, 2,2′,4′-trihydroxychalcone; Zer, zerumbone; IsoR, isorhamnetin; Chr, chrysoeriol.

**Table 1 antioxidants-09-01193-t001:** Effects of phytochemicals on autophagy flux and p62 expression in Caco-2 cells.

	Relative GFP/RFP Ratio	Relative p62 Levels
Control	1.00 ± 0.01	1.00 ± 0.00
Torin1	0.81 ± 0.02	0.70 ± 0.12
Zerumbone	0.88 ± 0.01	1.40 ± 0.15
Isorhamnetin	0.83 ± 0.02	0.83 ± 0.04
Chrysoeriol	0.90 ± 0.01	1.35 ± 0.22
2,2′,4′-Trihydroxy chalcone	0.88 ± 0.02	1.14 ± 0.13

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
