# Peer review of "Identification of Dietary Phytochemicals Capable of Enhancing the Autophagy Flux in HeLa and Caco-2 Human Cell Lines"

_antioxidants, 2020, doi:10.3390/antiox9121193_

Round 1

Reviewer 1 Report

Major:

  1. There is a lack of valid justification for conducting experiments on both types of cancer cell lines. Among the two cells, it would be better to focus on the results for one cell and to present the other cells as ancillary.
  2. In the title, it is recommended to modify the "cultured human cell line" to the specific cell name used mainly in the experiment.

Minor:

1.The figure 6 legend says "mTORC1 activity was assessed by determining the level of phospho-p70 S6K and 4EBP1 band shifts", but please correct it to match the description in the text.

2.The expression “The results of this study may provide fundamental knowledge for the development of possible preventive strategies for various intractable diseases.” needs correction. Because “various intractable diseases” is somewhat exaggerated.

3. Please correct typo errors. Some are there.

Author Response

Our responses to reviewer1’s comments

Reviewer1’s comments and suggestions are shown in italics.

Major:

  1. There is a lack of valid justification for conducting experiments on both types of cancer cell lines. Among the two cells, it would be better to focus on the results for one cell and to present the other cells as ancillary.

Response: Thank you very much for your important comment. As you pointed out, the justification for using two types of cancer cell lines might be unclear. We would, therefore, like to explain our rationale here. We first conducted the high-throughput experiments to screen the active phytochemicals and analyze the structure-activity relationship. To quantitatively measure the activities of many phytochemicals, we used the HeLa cell line since it has been well-established as a high-throughput screening system by Kaizuka et al. We subsequently assessed the availability and molecular mechanisms of the identified active compounds using human colon carcinoma Caco-2 cells as we assume that the gastrointestinal tract may serve as a crucial organ exposed to higher concentrations of dietary phytochemicals due to their reported poor absorption in vivo.

We have revised the unclear statements in the Abstract, Introduction, and Results sections, as follows:

Line 22-23 (ORIGINAL)

“Among them, isorhamnetin, chrysoeriol, 2,2’,4’-trihydroxychalcone, and zerumbone significantly enhanced autophagy flux in HeLa and Caco-2 cells.”

Line 22-23 (REVISED version)

“Among them, isorhamnetin, chrysoeriol, 2,2’,4’-trihydroxychalcone, and zerumbone enhanced autophagy flux in HeLa cells.”

Line 25-27 (ORIGINAL)

“Certain active compounds increased the expression of p62 protein, a typical substrate of autophagic proteolysis, indicating that phytochemicals act on p62 levels in autophagy-dependent and/or -independent manner.”

Line 26-29 (REVISED version)

“These active compounds were also effective in colon carcinoma Caco-2 cells, some of which increased the expression of p62 protein, a typical substrate of autophagic proteolysis, indicating that phytochemicals impact p62 levels in autophagy-dependent and/or -independent manners.”

Line 100-104 (ORIGINAL)

“In this study, we prepared a screening system capable of quantifying the cellular autophagy flux using a GFP-LC3-RFP-LC3DG probe and Operetta, a high-content imaging system. We then screened 67 types of phytochemicals, and identified four dietary components that enhance the autophagy flux in HeLa and Caco-2 cells. As for active compounds, structure-activity relationships and modes of action were further investigated.”

Line 103-107 (REVISED version)

“In this study, we prepared a screening system capable of quantifying cellular autophagy flux using HeLa cells expressing a GFP-LC3-RFP-LC3DG probe and Operetta, a high-content imaging system. We then screened 67 types of phytochemicals and identified four dietary components that enhance the autophagy flux. As for active compounds, autophagy-enhancing activities, and modes of action in the intestine were further investigated using the Caco-2 human colon carcinoma cell line.”

Line 310-311 (ORIGINAL)

“Next, we generated human colon carcinoma Caco-2 cells stably expressing the GFP-LC3-RFP probe to investigate the effects of phytochemicals on autophagy flux in human intestinal cells.”

Line 313-316 (REVISED version)

“The gastrointestinal tract is considered to be a crucial organ that is exposed to high concentrations of dietary phytochemicals [47]. Thus, we generated human colon carcinoma Caco-2 cells stably expressing the GFP-LC3-RFP probe to investigate the effects of phytochemicals on autophagy flux in human intestinal cells.”

Major:

  1. In the title, it is recommended to modify the "cultured human cell line" to the specific cell name used mainly in the experiment.

Response: Thank you very much for your valuable suggestion. Accordingly, we have modified the title, as follows.

Line 2-4 (ORIGINAL)

“Identification of dietary phytochemicals capable of enhancing the autophagy flux in cultured human cell lines”

Line 2-4 (REVISED version)

Identification of dietary phytochemicals capable of enhancing the autophagy flux in HeLa and Caco-2 human cell lines

Minor:

  1. The figure 6 legend says "mTORC1 activity was assessed by determining the level of phospho-p70 S6K and 4EBP1 band shifts", but please correct it to match the description in the text.

Response: Thank you very much for kindly pointing this out. We have modified this statement in the Results section as follows:

Line 347-349 (ORIGINAL)

“We further investigated whether mTORC1 signaling was influenced by these phytochemicals by using band shift assays to evaluate the phosphorylation status of the mTORC1 substrates, p70 S6K and 4EBP1.”

Line 351-354 (REVISED version)

“We further investigated whether mTORC1 signaling was influenced by these phytochemicals using an anti-phospho-p70 S6K (for detecting Thr389 phosphorylation) and an anti-4EBP1 antibody (for assessing its phosphorylation via band shift) to evaluate the phosphorylation status of the mTORC1 substrates, p70 S6K and 4EBP1.”

  1. The expression “The results of this study may provide fundamental knowledge for the development of possible preventive strategies for various intractable diseases.” needs correction. Because “various intractable diseases” is somewhat exaggerated.

Response: Thank you for this comment. Accordingly, we have revised the text in the Introduction and Conclusions as follows:

Line 104-105 (ORIGINAL)

“The results of this study may provide fundamental knowledge for the development of possible preventive strategies for various intractable diseases.”

Line 108-110 (REVISED version)

“The results of this study may provide fundamental insights into the development of possible preventive strategies for various diseases, such as neurodegenerative and lifestyle-related diseases associated with autophagy dysfunction.”

Line 465-466 (ORIGINAL)

“These findings could serve to inform the development of novel preventive strategies against various intractable diseases.”

Line 475-478 (REVISED version)

“These findings could serve to inform the development of novel preventive strategies against various diseases, such as neurodegenerative and lifestyle-related diseases associated with autophagy dysfunction.”

  1. Please correct typo errors. Some are there.

Response: Thank you for raising this issue. We have corrected the typographical errors throughout the manuscript and have had the language re-checked and polished by Editage (www.editage.com).

Reviewer 2 Report

Manuscript “Identification of dietary phytochemicals capable of  enhancing the autophagy flux in cultured human cell  lines” by  Kohta Ohnishi et al. presents data about effect of   67 diferent dietary phytochemicals in 2 cell lines Hela and  CaCo2 . Authors prepared system for detection based on stable expression of  GFP-LC3-RFP-LC3DG probe for quantitative assessment of autophagic flux. Four detected effective compounds were further analysed for structure –activity relationship, and their effect on p62 protein level and mTOR signalling  was assessed.

Manuscript is well written with minor  issues to address:

  • Suplemental figure was not available on link
  • Since manuscript is also addressing important questions of (  p62 level) measurement, bplease include one  figure showing scheme of autophagy and signalling pathways  with key   players  addressed in introduction and  discussion., and possible pitfalls.

411” Table 1-effects of phytochemiclas on autophagy flux and p62 expression”  in  ??? name the model….

Round 2

Reviewer 1 Report

Please correct simple typing errors all over the sentence.